# Update on the Role and Regulatory Mechanism of Extracellular Matrix in the Pathogenesis of Uterine Fibroids

**DOI:** 10.3390/ijms24065778

**Published:** 2023-03-17

**Authors:** Qiwei Yang, Ayman Al-Hendy

**Affiliations:** Department of Obstetrics and Gynecology, University of Chicago, Chicago, IL 60637, USA; aalhendy@bsd.uchicago.edu

**Keywords:** uterine fibroids, extracellular matrix, stiffness, mechanotransduction, signaling pathways, treatment options

## Abstract

Uterine fibroids (UFs), also known as leiomyomas, are benign tumors of the myometrium affecting over 70% of women worldwide, particularly women of color. Although benign, UFs are associated with significant morbidity; they are the primary indication for hysterectomy and a major source of gynecologic and reproductive dysfunction, ranging from menorrhagia and pelvic pain to infertility, recurrent miscarriage, and preterm labor. So far, the molecular mechanisms underlying the pathogenesis of UFs are still quite limited. A knowledge gap needs to be filled to help develop novel strategies that will ultimately facilitate the development of therapies and improve UF patient outcomes. Excessive ECM accumulation and aberrant remodeling are crucial for fibrotic diseases and excessive ECM deposition is the central characteristics of UFs. This review summarizes the recent progress of ascertaining the biological functions and regulatory mechanisms in UFs, from the perspective of factors regulating ECM production, ECM-mediated signaling, and pharmacological drugs targeting ECM accumulation. In addition, we provide the current state of knowledge by discussing the molecular mechanisms underlying the regulation and emerging role of the extracellular matrix in the pathogenesis of UFs and in applications. Comprehensive and deeper insights into ECM-mediated alterations and interactions in cellular events will help develop novel strategies to treat patients with this common tumor.

## 1. Introduction

Uterine fibroids (UFs), also known as leiomyomas or myomas, are benign tumors of the myometrium affecting over 70% of women worldwide, particularly women of color. Although benign, UFs are associated with significant morbidity; they are the primary indication for hysterectomy and a major source of gynecologic and reproductive dysfunction, ranging from menorrhagia and pelvic pain to infertility, recurrent miscarriage, and preterm labor [1,2,3]. Each year, approximately 300,000 myomectomies and 200,000 hysterectomies are performed in the United States to remove either leiomyoma tumors or the whole uterus [4,5]. Accordingly, the annual USA healthcare costs associated with UFs have been estimated at ~34 billion USD. Therefore, UFs represent significant societal health and financial burdens.

Several risk factors have been shown to impact UF pathogenesis and are associated with a higher probability of UF formation and development. These factors include race, age, parity, family history, food additives, obesity, vitamin D deficiency, and endocrine-disrupting chemical exposure [1,6]. These risk factors affect several key pathways, including inflammation [7,8,9], DNA damage repair pathway, β-catenin pathway, and genetic instability, among others, leading to the pathogenesis of UFs [10,11,12]. Despite the importance to women’s health, there are currently no UF-specific therapeutics because UFs are heterogeneous in composition and size among women, even within the same individual, and vary in number between individuals [13,14,15,16]. These irregularities highlight the challenge of preventing UFs and treating patients with UFs. Moreover, our understanding of the origin and pathogenesis of UFs continues to evolve.

## 2. Extracellular Matrix and Regulation in Uterine Fibroids

Excessive extracellular matrix (ECM) accumulation and aberrant remodeling are crucial for fibrotic diseases. An increased stiffness characterizes the fibrotic microenvironment, and this rigidity is associated with disease progression. The mechanical network of fibrotic ECM is regulated by ECM-degrading enzymes called matrix metalloproteinases (MMPs). MMPs are commonly classified on the basis of their substrates and the organization of their structural domains into collagenases, gelatinases, stromelysins, matrilysins, membrane-type (MT)-MMPs, and other MMPs. MMPs are often secreted in an inactive pro-MMP form, which is cleaved to the active form by various proteinases, including other MMPs. MMPs can be regulated by endogenous tissue inhibitors of metalloproteinases (TIMPs), and the MMP/TIMP ratio often determines the extent of ECM protein degradation and tissue remodeling [17]. UFs are characterized by the excessive deposition of ECM proteins, such as collagens, fibronectin, and proteoglycans, representing fibrosis [18,19,20]. In addition to MMPs and TIMPs, several factors impact ECM accumulation and deposition in UFs.

### 2.1. ECM and Hormones

UFs are considered hormone-dependent tumors, based on their association with reproductive age. Estrogen and progesterone are considered the principal promoters of UF growth [1,21,22]. Estrogen-related signaling impacts the biological process via genomic and nongenomic mechanisms. In addition, estrogen is capable of inducing the expression of the progesterone receptor (PR), stimulating progesterone-regulated signaling. Estrogen exerts multiple stimulatory actions on its target cells and accelerates collagen biosynthesis [23,24]. Three-dimensional UF cell cultures exposed to estrogen and medroxyprogesterone acetate (MPA) increased in their expression of collagen I and fibronectin [25].

FK506-binding protein 51 (FKBP51) is known as a chaperone that regulates the responsiveness of steroid hormone receptors. It was linked to several intracellular pathways related to tumorigenesis and chemoresistance [26,27]. FKBP51 was shown to bind PR, glucocorticoid receptor (GR), and androgen receptor (AR) to coregulate their transcriptional activity. It was reported that FKBP51 expression is higher in UFs compared to myometrial tissues. The knockdown of FKBP51 displayed decreased mRNA levels of ECM, TIMP1, and TIMP3, and reduced cell proliferation [28].

Prolactin is a hormone responsible for lactation and has been reported to be present and functional in UFs [29]. Prolactin strongly activates STAT5 and MAPK signaling in rat and human myometrial cell lines. Moreover, Prolactin produced from UFs may stimulate the transdifferentiation of the myometrium (MM) cells to myofibroblasts, contributing to the fibrotic nature of UFs [29].

### 2.2. ECM and Growth Factors

The ECM acts as a reservoir of profibrotic growth factors and enhances their activity by increasing their stability and prolonging signaling duration. Therefore, a better understanding of ECM composition and metabolism in UFs is critical for developing new therapeutics for UFs. In addition, several growth factors have been shown to trigger the ECM accumulation in UFs, including transforming growth factor-β, activin-A, and platelet-derived growth factor [24].

TGF-β initiates the cascade of signal transduction that elicits biological actions on responding cells via receptors on the plasma membrane. The central mechanism of signal transduction by the TGF-β family receptors follows a well-characterized process of interactions and receptor-mediated phosphorylation. Upon ligand binding and the following cascade steps, the trimeric complex (SMAD2, 3, 4) translocates into the nucleus and associates with high-affinity DNA binding transcription factors and chromatin remodeling proteins, thereby positively or negatively regulating the transcription of the TGF-β-responsive genes [30]. Numerous studies demonstrated that TGF-β signaling plays an important role in the pathogenesis of UFs. Abnormal ECM accumulation and deposition have been shown to be associated with the activation of TGF-β signaling in UFs. Several studies have demonstrated that TGF-β3 stimulates the production and secretion of ECM macromolecules and alters the expression of MMP members [24,31].

Activin-A is a member of the TGF-β superfamily, a large family of over 30 structurally related proteins. Activin A has been recognized as a multifunctional cytokine expressed in a wide range of tissues and cells, and growing evidence implicates activin A in the pathogenesis of UFs. In vitro studies demonstrated that Activin A promoted cell proliferation and increased ECM protein accumulation via p38 MAPK signaling in immortalized UF cells [32,33]. In addition, activin A significantly increased mRNA expression of FN, collagen 1A1, and versican in primary UF cells concomitantly with the activation of Smad-2/3 signaling, but not with changes in ERK and P38 signaling [34].

The platelet-derived growth factor (PDGF) family belongs to the growth factor systems. Its dysregulation is involved in a wide array of pathological conditions, such as fibrosis, neurological disorders, atherosclerosis, and tumorigenesis [35]. The PDGF family consists of four members: PDGF-A, PDGF-B, PDGF-C, and PDGF-D. Members of the PDGF family bind to and signal through the PDGF tyrosine kinase receptors with an extracellular ligand-binding domain and an intracellular tyrosine kinase domain. Upon ligand binding, the receptor dimerization results in receptor autophosphorylation on tyrosine residues. Autophosphorylation further activates the receptor kinase and docking sites for downstream signaling molecules and the modulation of different pathways [36]. PDGF is upregulated in about 80% of UFs compared to adjacent myometrial tissues [37] and can promote the growth of myometrial and UF-derived smooth muscle cells, which is one of the main cell populations contributing to ECM production and secretion. In addition, PDGF-C prolongs the survival of UF-derived SMCs in Matrigel plugs implemented subcutaneously in immunocompromised mice. Furthermore, PDGF can increase the collagen levels in both myometrial and UF cells [38].

### 2.3. ECM and Cytokines

Cytokines are small proteins with characteristics of intercellular messengers, which have a complex regulatory influence on inflammation and immunity. In addition, cytokines play an important role in many other biological processes, including tumorigenesis [39]. Proinflammatory cytokines have been shown to cause potent and consistent changes in ECM expression in many cell types [40,41]. Several cytokines have been implicated in the development of UFs [42]. Tumor necrosis factor-α (TNF-α) is a cell-signaling protein involved in systemic inflammation and is one of the cytokines responsible for the acute phase reaction. It was reported that TNF-α serum levels in women with clinically symptomatic UFs were significantly higher than in the control group [43]. TNF-a can increase the expression of activin-A, the ECM inducer, in UF and MM cells, suggesting that TNF-a may increase the deposition of ECM, leading to UF pathogenesis [24,44]. Further investigation of TNF-a-induced alterations in ECM production and components will help better understand the cytokine role in the ECM-mediated signal pathway in UFs.

### 2.4. Cell Types Contributing to ECM Production in Uterine Fibroids

UF heterogeneity exists at many levels, including etiology, clinical symptoms, and pathogenesis, which have significant ramifications for research design and therapeutic decisions [13,45]. The ECM forms a milieu surrounding cells that reciprocally influence cellular function to modulate diverse fundamental aspects of cell biology [46].

Intracellular heterogeneity is present in UFs with multiple cellular compositions [47]. Among them, SMCs and fibroblast populations are dominant in contributing to ECM secretion and participate in the collagen signaling network in the MED12-variant-positive UFs compared to the MM tissues [48]. Accordingly, increased SMC and fibroblast proliferation in UFs is correlated to enhanced ECM accumulation, the characteristic feature of UFs [37,49]. In addition, there was a significant increase in UF cells when cocultured with UF-derived fibroblasts. UF-derived fibroblasts can stimulate the production of collagen type I in the medium cocultured with UF cells [50].

## 3. ECM and Downstream Signaling

Several studies have demonstrated that ECM contributes to mechanotransduction in UFs. Excessive ECM deposition and abnormal remodeling can regulate downstream signaling in UF cells.

### 3.1. Stiff ECM and Progesterone Receptor Signaling

The uterine myometrium consists of various cell types and is interspersed with interstitial ECM. Cells grown on mechanically stiff ECM substrate promote progesterone receptor activation via MEK1/2 and AKAP/RhoA/ROCK signaling pathways. Accordingly, UF cells exhibited higher RNA levels of collagen I grown on stiff collagen I-coated- plates compared to soft collagen I-coated plates in response to progesterone treatment [51].

### 3.2. The Effect of Collagen Cross-Linking on Proliferation and Resistance to MMP Proteolytic Degradation

Collagen, as one of the major ECM components, contains various types of collagens. The amount of collagen can influence the conversion of signals into chemical changes, and intramolecular and intermolecular Lysyl oxidase cross-links can form the mature collagen fibrils that impact these functional signals [20]. The expression levels of collagen cross-linking enzymes lysyl hydroxylases (LH) and lysyl oxidases (LOX) are higher in UFs than MM tissues linked to increased collagen cross-links and resistance to MMP proteolytic degradation, highlighting the role of cross-linking in ECM remodeling. In addition, increased collagen cross-links positively correlated to the UF size and cell-proliferation-related markers [52].

### 3.3. ECM Stiffness and Other Signaling

It has been reported that ECM stiffness impacts several signaling pathways in UFs. The Wnt signaling pathway is involved in various cellular processes such as embryogenesis, tissue renewal, cell proliferation, differentiation, and tumorigenesis [53,54,55,56]. In canonical Wnt-on signaling, β-catenin avoids destruction in the cytoplasm and translocates into the nucleus. Subsequently, the nuclear β-catenin binds to the TCF/LEF transcription factors and accelerates β-catenin-regulated gene expression. In canonical Wnt-off signaling, a combination of AXIN and APC allows GSK3β to phosphorylate β-catenin and targets it for proteasomal degradation. Wnt/β-catenin signaling is abnormally activated in UFs compared to MM [57], and the mislocalization of β-catenin is correlated to the UF phenotype [58]. Moreover, estrogen triggers β-catenin nuclear translocation and enhanced β-catenin-responsive gene expression in human UF cells. An increase in ECM stiffness occurred when primary UF cells were grown on hydrogels of known stiffness and triggered the upregulation of β-catenin expression in UF cells [57].

The Hippo signaling pathway is highly conserved and plays a critical role in tumorigenesis. It is characterized by the phosphorylation of YAP1 and TAZ. Several factors can regulate the localization of Yes-associated protein (YAP), therefore modulating the downstream signaling. Hippo pathways can integrate mechanical signaling with cell growth. YAP is a key transcriptional coactivator of the Hippo pathway and acts as a sensor and regulator of a wide range of physical and mechanical stresses, including the stiffness of the ECM [59]. The increased ECM stiffness activates YAP, promoting cell growth and fibrotic gene transcription. On the other hand, in the soft matrix, YAP becomes phosphorylated or degraded in the cytoplasm. When YAP, a suppressor of the YAP-TEAD complex, is phosphorylated, it becomes transcriptionally inactive. A study by Purdy et al. showed that decreased substrate stiffness reduced YAP/TAZ nuclear localization in both the MM and UFs [60]. The pharmacological inhibition of YAP with verteporfin reduced key targets of fibrosis and genes involved in mechanotransduction in UF cells. In addition, the antifibrotic drug nintedanib inhibited YAP and showed antifibrotic effects on UF cells with decreased fibronectin levels [61].

## 4. Targeting ECM

### 4.1. Targeting Hormone-Dependent Growth with ECM Changes

The significant variations in the quantity and distribution of the ECM also create variation in the degree of UF tissue stiffness from soft to firm to solid to hard, with a major impact on the gene expression and biology of these tumor lesions. UF growth is regulated by hormone levels in the body (estrogen and progesterone are both promoters of UF growth), providing the basis for new therapy developments, with others in the pipeline. Hormonal contraception containing estrogen and progestin effectively prevents unintended pregnancy and lowers the risk of some female cancers. However, it may increase other risks and cause some side effects in women. Current pharmacological treatment options for UFs aim to halt their hormonal-dependent growth using gonadotropin-releasing hormone (GnRH) analogs (agonists and antagonists). For example, Bozzini et al. [62] identified significant positive correlations between UF volume reduction and collagen content, progesterone receptors, and blood supply, and negative correlations with estrogen receptors after treatment with GnRH agonists (goserelin). The following efforts aimed to target the GnRH to improve outcomes and reduce hypoestrogenic side effects, such as bone loss, in clinical trials. Such concerns led to the hypothesis that one might maximize the benefits of blocking UF-associated menstrual bleeding and minimize the hypoestrogenic side effect. This idea led to the design of a series of studies that demonstrate that new-generation oral GnRH antagonists, including elagolix [63], relugolix [64], and linzagolix [65], are promising, well tolerated, and noninvasive options, and offer potential prospects for the future therapy of UFs. These clinical trial studies support the hormone threshold approach to increase the quality of life of premenopausal women with clinically significant UFs. Notably, a recent study demonstrated that the new-generation GnRH antagonists relugolix and elagolix decreased the production of the ECM component fibronectin in UF cells, accompanied by MAPK inhibition [66].

### 4.2. Collagenase

Since UFs are collagen-rich, fibrotic tumors, the digestion of collagen would degrade the ECM, altering the stiffness. The injection of a highly purified form of collagenase Clostridium histolyticum into UFs resulted in a remarkable reduction in fibrosis and stiffness, concomitantly with a decrease in cell proliferation markers [61,67]. 

### 4.3. Vitamin D and ECM

Vitamin D forms a complex with a specific receptor named the vitamin D receptor (VDR) to mediate its pleiotropic functions via steroid transcriptional mechanisms [1]. Vitamin D deficiency is one of the critical risk factors for UF pathogenesis [68,69]. Our previous studies demonstrated that vitamin D3 reduced UF cell proliferation in vitro and UF growth in vivo animal models. Vitamin D3 decreased key ECM components, including collagen type I, fibronectin, and proteoglycans [70]. In addition, Vitamin D3 altered the expression and activities of matrix metalloproteinase 2 and 9 in human UF cells, indicating its role in ECM remodeling. These results underscore the potential role of vitamin D3 in an effective, safe, nonsurgical medical treatment option for UFs [71]. The key ECM inducers and relevant pathways and inhibitors are summarized in Table 1 and Figure 1.

### 4.4. Epigenetic-Mediated ECM Changes

HDACs are a class of enzymes that remove acetyl groups from an N-acetyl lysine amino acid on a histone, allowing the histones to wrap the DNA more tightly, thereby regulating gene expression. In addition, nonhistone proteins are also targeted by HATs and HDACs, modulating various biological events. Studies have demonstrated that abnormal HDAC signaling contributes to many diseases, including tumorigenesis [72,73,74]. In UFs, HDAC expression and activity are abnormally upregulated compared to MM tissues [58,75]. In one study, the HDAC inhibitor suberoylanilide hydroxamic acid (SAHA) selectively inhibited UF growth but not MM cells [75]. In addition, SAHA significantly inhibited ECM production, including FN and collagen I, accompanied by a decrease in TGF-β3, MMP9, and cell-cycle-related markers (C-MYC and CCND1) [75]. Resveratrol is a polyphenolic phytoalexin found in peanuts, grapes, and other plants with characteristics of antioxidant and anti-HDAC activity [76]. Resveratrol treatment suppressed UF growth in a UF xenograft model and decreased the expression levels of collagen 1, a-SMA, and FN, demonstrating the potential role of the antifibrotic effect of HDAC inhibition [77].

Bromodomains (BRDs) are evolutionarily conserved protein–protein interaction modules that selectively recognize and bind to acetylated lysine residues—particularly in histones—and thereby have important roles in regulating gene expression. Bromodomains (BRD)-containing proteins are responsible for transducing regulatory signals from acetylated lysine residues into various biological phenotypes. BRD proteins can have various functions via multiple gene regulatory mechanisms, including chromatin remodeling, histone modification, histone recognition, and scaffolding, and various activities in transcriptional coregulation; therefore, they can lead to the alteration of gene transcription [78,79]. The BET (bromodomain and extraterminal domain) family of proteins, consisting of BRD2, BRD3, BRD4, and testis-specific BRDT, are widely acknowledged as major transcriptional regulators in biology. They are characterized by two tandem bromodomains (BRDs) that bind to lysine-acetylated histones and transcription factors, recruit transcription factors and coactivators to target gene sites, and activate RNA polymerase II machinery for transcriptional elongation. The pharmacological inhibition of BET proteins with BRD inhibitors has been shown as a promising therapeutic strategy for treating many human diseases, including cancer [80]. BRD4 expression is abnormally upregulated in breast cancer tissues and cells, and the targeted inhibition of BRD4 by pharmacological inhibitors and the genetic loss of function analysis have been shown to significantly inhibit the malignancy of breast cancer cells and concomitantly decrease the fibronectin protein levels [81]. Attention has now expanded to the biological role of non-BET BRD proteins. Among them, non-BET BRD-containing protein 9 (BRD9) is associated with various diseases, including tumorigenesis. Our recent studies demonstrated that the pharmacological inhibition of non-BET BRD9 suppressed the UF phenotype with decreased ECM proteins, including fibronectin in UF cells [82], reinforcing the view that BRD proteins may be involved in the pathogenesis of UFs.

DNA methylation is one of the common epigenetic regulation mechanisms in eukaryotes. The hypermethylation of the CpG island in the promoter region generally results in the repression of gene expression, while hypomethylation leads to active transcription. Cytosine methylation is catalyzed by specific DNA methyltransferases (DNMTs) that transfer a methyl group from the donor S-adenosyl methionine to the 5′-position of the pyrimidinic ring. The cytosine methylation can be oxidized by the TET dioxygenases to ultimately cause DNA demethylation [83]. Aberrant DNA methylation dynamics occur in many diseases, including tumorigenesis. In UFs, abnormal DNA methylation is involved in the pathogenesis of UFs [83,84,85]. DNMT activity was shown to be upregulated, indicating an important role of DNA methylation in UFs [86]. Targeting DNMT with 5-aza-2′-deoxycytidine, the DNMT inhibitor, reduced the collagen I levels, concomitantly with a decrease in the wnt/βcatenin pathway, MMP7, and c-MYC in UF cells [86] (Figure 2).

MicroRNAs (or miRNAs) are small, noncoding RNAs (∼22 nt long) that regulate post-transcriptional gene expression. By binding to the target mRNAs’ 3′-UTR (untranslated region), miRNAs prevent protein production by inducing mRNA degradation and/or directly repressing translation. A number of studies demonstrated that miRNAs play an important role in the pathogenesis of UFs [9,87,88,89,90]. The expression levels of the mir-29 family, including mir-29a, mir-29b, and mir-29c, were downregulated in UFs compared to the MM tissues. The overexpression of each mir-29 family member reduced the ECM production in UF cells [91,92,93,94]. Another microRNA, miR-139-5p, regulated fibrotic potentials via the modulation of collagen type I and phosphorylated p38 MAPK in UFs [95]. The expression of miR-129 was also lower in UFs. The targeted inhibition of mir-129 modulated the accumulation of ECM, accompanied by a decrease in cell proliferation and TET1 expression [96].

These studies suggested that the epigenetic targeting of UF could reverse the UF phenotype via decreasing ECM composition and accumulation, one of the key hallmarks of UFs (Figure 2).

### 4.5. ECM and Other Inhibitors

Several other factors have been shown to regulate ECM production and stiffness (Table 1). Simvastatin is used to help lower “bad” cholesterol and fats and raise “good” cholesterol, and exhibits anti-UF effects both in vitro and in vivo [97,98]. The treatment of UF cells with simvastatin decreased the production of collagen, fibronectin, versican, and brevican and reduced the levels of mechanical signaling proteins involved in β1 integrin downstream signaling [99,100].

Ulipristal acetate as a selective progesterone receptor modulator showed a decrease in the total volume of UFs and an improvement in quality-of-life measures [101,102,103]. In addition, in a randomized control study, the administration of ulipristal acetate decreased fibronectin, and versican, concomitantly altering the expression levels of MMP2 and MMP9 [104].

**Table 1 ijms-24-05778-t001:** Summarized ECM research in uterine fibroids since 2018.

Inducers/Inhibitors	Biological Samples	Changes in ECM	Approach	Co-Changes	Publication Time	References
Activin A	UF cell line	Increase in ECM accumulation	WB	p38 MAPK	July 2021	[33]
Activin A	UF cell line	Excessive ECM	WB, qPCR, and IC	p38 MAPK	October 2018	[32]
Butylated hydroxytoluene	ELT-3 UF cell line	Increase in CO1A1	WB, IF	PI3K/AKT and MAPK	August 2021	[105]
Leptin	UF cell line	Increase in ECM formation		JAK2/STAT3 and MAPK/ERK	May 2022	[106]
TBBPA	3D human UF spheroids	Increase in collagen and fibrosis	Masson’s trichrome stain, PCR, and light microscopy	TGF-β signaling	February 2022	[107]
Cadmium prolonged exposure	UF cells	Decrease in collagens, FNs, laminins; increase in MMPs	IF and MMP antibody array	TGF-β signaling, cell proliferation, and migration	August 2021	[108]
Collagenase Clostridium histolyticum	UF tissues	Decrease in ECM stiffness	Rheometry	Cell proliferation and Hippo signaling	July 2021	[61]
CRSR	UF rat model	Decrease in ECM deposition	Microarray	MAPK, PPAR, Notch, and TGF-β	May 2019	[109]
Decrease in FKBP51	UF cells	Decrease in ECM formation	qPCR	Cell survival and proliferation	June 2022	[28]
Fucoidan	ELT-3, human UF cells, xenograft model	Decrease in FN and COL1A1	WB	TGF-β3 signaling	September 2018	[110]
Isoliquiritigenin	ELT3, UtSMC, uterine myometrium hyperplasia mouse model	Decrease in ECM accumulation	WB and IHC	MMP, ERK1/2, p38, and JNK	August 2019	[111]
miR-139-5p	UF cell line	Decrease in contractility of the ECM	Migration, collagen gel contraction, and wound healing	p38 MAPK	August 2021	[95]
miR-21a-5p	UF and MM cell lines	FN, Collagen 1A1, CTGF, versican, and DPT	qPCR and collagen assay	TGF-β3 and MMPs	May 2018	[112]
Relugolix and elagolix	UF cells	Decrease in collagen 1A1, FN, and versican		pERK/ERK	August 2022	[66]
Resveratrol	Xenograft and UF primary cells	Decrease in collagen I, FN, and a-SMA	WB, qPCR	Proliferation and apoptosis	April 2019	[77]
Simvastatin	UF SCs	Decrease in collagen I and fibronectin	WB	TGF-β and β-catenin signaling	February 2022	[97]
Simvastatin	UF cell lines and primary cells, 2D and 3D	Decrease in collagen I, collagen III, FN, versican, and brevican	WB and IHC	Apoptosis	December 2018	[99]
S1P	UF and myometrium cells	Decrease in FN and collagen 1A1	qPCR	Decrease in activin A	June 2021	[113]
Butylated hydroxytoluene	UF primary cells	Decrease in collagen I and fibronectin	WB	Cell cycle and TGF-β3 signaling	February 2022	[75]
Ultrasound-guided collagenase injection	UF tissues	Decrease in content, density, and fibers of ECM	Masson’s trichrome stain, second harmonic generation, and Picrosirius stain	Decrease in UF-related pain	September 2021	[114]
Ulipristal acetate	UF patients	Decrease in versican and FN1	Masson trichrome staining and IHC	Decrease in UF size	February 2018	[104]
Vitamin D	Xenograft	Decrease in collagen I and plasminogen activator inhibitor 1	WB	TGF-β3 signaling	January 2020	[115]

## 5. Conclusions and Future Perspectives

Although the heterogeneity of UFs is present and underscores the challenge for the consideration of basic, translational, or clinical studies or standard clinical practice, considerable progress has been made in recent years to study the role and mechanisms underlying the ECM-mediated cellular and molecular events which contribute to the pathogenesis of UFs. ECM deposition and remodeling can be regulated via hormones, growth factors, cytokines, and MMPs. Excessive ECM can induce mechanotransduction through integrin activation and increase tissue stiffness by altering bidirectional signaling, which leads to UF progression. Pharmacological drugs can suppress the UF phenotype by inhibiting ECM production via multiple mechanisms, including epigenetics (Figure 2). However, several aspects need to be further elucidated, including (1) characterizing the functional interaction of ECM-secreted cells with other types of cells, (2) elucidating the impact of ECM on the microenvironment, (3) characterizing the ECM-mediated signaling via its interactions with other biological pathways via autocrine and or paracrine mechanisms, and (4) investigating the molecular and functional structure of ECM that regulates mechanotransduction and stiffness in both UFs and the at-risk MM. The deep mechanistic and functional investigation will help us better understand the regulatory mechanism of ECM-secreted cell types, cell–cell interaction, and environmental impacts on the risk and development of UFs. The clinical application of ECM/stiffness measurement and molecular inhibitors targeting the ECM might provide a promising option for the personalized treatment of patients with UFs.

## Figures and Tables

**Figure 1 ijms-24-05778-f001:**
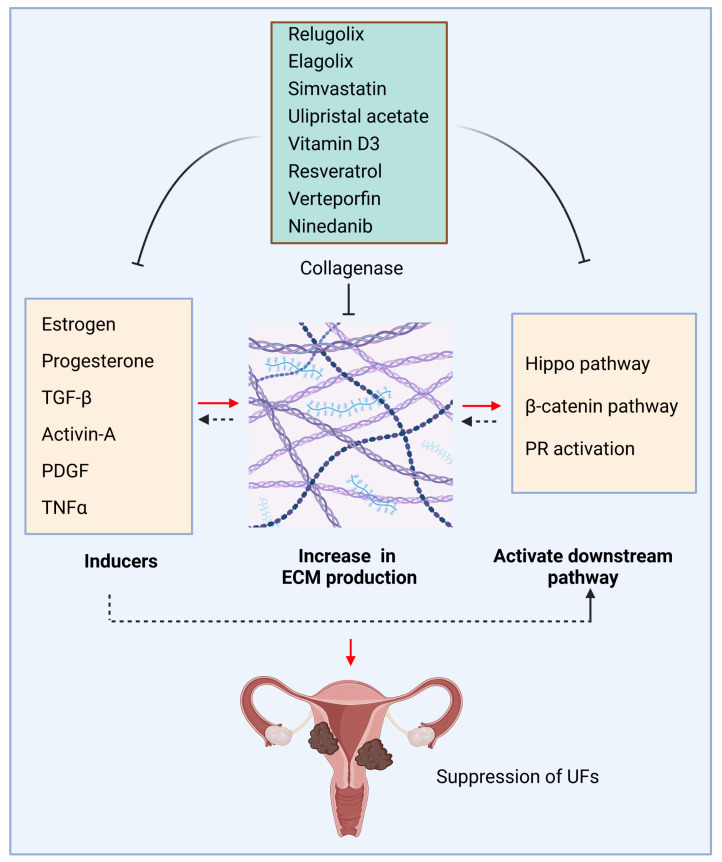
ECM production and inhibition in uterine fibroids. ECM inducers, including hormones, cytokines, and growth factors, trigger the deposition and accumulation of ECM, activating downstream pathways and creating a pathological loop to promote the pathogenesis of UFs. Conversely, targeted inhibition of ECM inducers and relevant pathways decreases the production of ECM, concomitantly with the changes in mechanotransduction and relevant signaling, which contribute to the development and progression of UFs. Black dotted arrow: need further investigation; This figure was created using BioRender software. PR: progesterone receptor.

**Figure 2 ijms-24-05778-f002:**
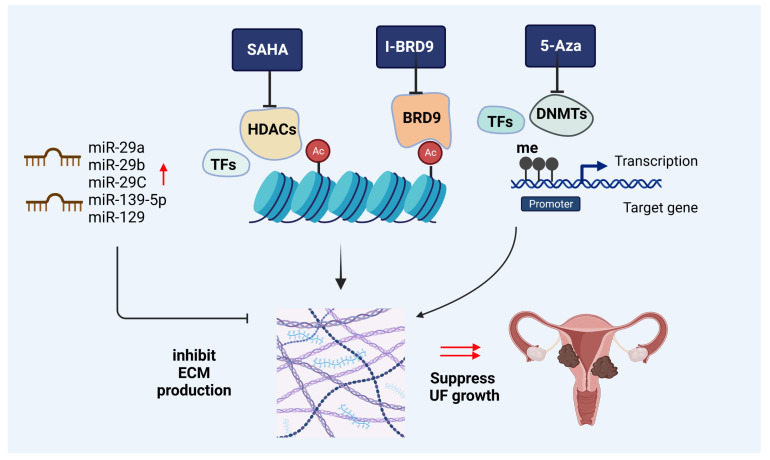
Epigenetic-mediated targeting on ECM production. Targeted inhibition of HDACs, BRDs, and DNMTs and overexpression of targeted microRNAs can decrease the ECM deposition and accumulation in UF cells, leading to inhibited UF growth. Red up-arrow: overexpression of the microRNAs, which lead to the inhibition of ECM production; black ┴: inhibition; HDACs: histone deacetylases; BRD9: bromodomain-containing protein 9; DNMTs: DNA methyltransferases; ECM: extracellular matrix; TFs: transcription factors; SAHA: suberoylanilide hydroxamic acid (HDAC inhibitor); I-BRD9: BRD9 inhibitor; 5-Aza: 5-Aza-2-deoxycytidine (DNMT inhibitor). This figure was created using BioRender software.

## Data Availability

Not applicable.

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
