# Peer review of "Update on the Role and Regulatory Mechanism of Extracellular Matrix in the Pathogenesis of Uterine Fibroids"

_ijms, 2023, doi:10.3390/ijms24065778_

Round 1
Reviewer 1 Report
In this manuscript, Yang and Al-Hendy summarize the role and regulatory mechanism of extracellular matrix in the pathogenesis of uterine fibroids. Since leiomyomas cause several gynecological and reproductive problems although they are benign, it is crucial to be able to prevent it to improve the quality of life. In this aspect, this review gives important insights, specifically about the interaction of ECM with UFs. They first focused on different aspects of ECM and the regulators of it. Then, they summarize the studies to revert the effects of ECM overproduction on the progression of UFs. There is just grammatical errors in the sections beginning with 3.1 (3.2, 3.3, 4.4 and so on) that sohuld be corrected.
Author Response
Thanks very much for your comments. Per your suggestion, the grammatical errors have been corrected and the relevant changes are highlighted in red.
Reviewer 2 Report
The paper can be considered an interesting work. Uterine fibroids are commons during reproductive age and could be a possible cause of infertility and/or miscarriage. So I think it's crucial to understand the pathophysiology of leyomioma in order to try to find a good strategy for treatment in symptomatic women in order to avoid/retrd any possibile surgical treatment.
About the possible hormone impact, I suggest to report some comments about a possible role of hormonal contraception.
Author Response
Thanks very much for your suggestion. We have included some comments about the possible role of hormonal contraception on page 6-7, highlighted in red.